# Influence of Top Seal Damage on Contact Seal in Ram Blowout Preventer

**DOI:** 10.3390/ma16093413

**Published:** 2023-04-27

**Authors:** Shiqiang Wang, Laibin Zhang, Jiamin Yu, Jianchun Fan

**Affiliations:** 1College of Safety and Ocean Engineering, China University of Petroleum (Beijing), Beijing 100100, China; 2Research Institute of Safety, Environmental Protection and Quality Supervision and Inspection, Chuanqing Drilling Engineering Co., Ltd., Guanghan 618300, China

**Keywords:** ram blowout preventer, top seal, corrosion defect, wear defect, sealing performance

## Abstract

Top seal failure of ram blowout preventer (BOP) is one of the main factors leading to well control risk. The constitutive model and parameters of nitrile butadiene rubber (NBR) were optimized by compression and tensile tests, and the failure analysis model of the contact seal of the ram BOP top seal was built. The nonlinear contact mechanical behavior of the connection part of the BOP top seal was analyzed by the finite element method. Then, the influence of corrosion and wear defects at the top seal position of the 2FZ35-70 BOP under rated working pressure on the contact seal were studied, and the results showed that the overall contact pressure distribution of the top seal corrosion defects was uniform, the local contact pressure of the corrosion pit edge increased, and the top contact pressure decreased. The overall contact pressure of the wear defect of the top seal decreased linearly, the contact pressure at the maximum depth of the wear defect was the smallest, and the contact pressure gradually decreased to both sides. Ultimately, to guarantee the safety and reliability of the ram BOP, it is suggested that the acceptable depths of the seal corrosion pit and the wear at the top of the ram BOP are 4.0 mm and 0.2 mm, respectively, thus the reliability evaluation problem of the quantitative seal of the ram BOP top seal is solved.

## 1. Introduction

With the increase in global exploration and development of oil and gas resources, drilling depths have been ever-increasing in recent years. Hence, the number of deep, ultradeep, and highly deviated wells has inevitably increased [1]. Due to the severe and complex formation conditions faced by drilling operations, it is increasingly difficult and costly to maintain pressure balances in wells and prevent overflows. Major accidents such as blowouts can cause serious casualties, property losses, and environmental pollution [2,3]. For example, on 20 April 2010, the Deepwater Horizon blowout and explosion caused the world’s largest offshore oil spill to date, killing 11 people and spewing 4.9 million barrels of crude oil into the Gulf of Mexico [4]. One of the key causes of the blowout was the failure of the BOP. Therefore, to effectively avoid accidents during drilling, it is necessary to place further requirements on the reliability and safety of BOPs [5].

A ram BOP is one of the key pieces of equipment for oil and gas well pressure control. It is mainly used to close the annular space between the casing and pipe string to achieve the pressure balance in the well. Therefore, the tightness of the BOP under pressure is an important indicator of well control equipment [6]. To effectively block the high-pressure liquid inside the cavity, four seals are needed inside the ram BOP, namely, the seal between the ram top and the shell, the seal at the front of the gate to the string, the seal between the shell and the side door, and the seal between the piston rod and the side door. They work simultaneously to seal the wellhead annulus. However, the shell of a ram BOP can become fouled by mud, oil, and sand, which can affect the flatness and roughness of the shell, and easily cause seal failure [7].

In recent years, the sealing performance of rubber materials has often been analyzed by finite element theory. The representative research conclusions show that when a BOP seal annulus is sealed, the parts where the rubber core is in contact with the shell and piston are in a stress concentration position, and these positions of the rubber core are easily damaged due to the high stress [8,9,10]. Dong et al. conducted a study on the influence of a pipe joint passing through an annular BOP rubber core and its sealing performance; their results showed that the significant reduction in contact pressure caused by the pipe joint passing through the rubber core was one of the main reasons for rubber core seal failure [11].

The contact pressure between the rotary blowout preventer (RBOP) core and the drill pipe is affected by the well pressure, tripping speed, friction coefficient, and other factors. Gang et al. concluded that an increase in well pressure and friction coefficient would increase the contact pressure by analyzing the distribution law of contact pressure during the operation of the rubber sealing surface of the RBOP, subsequently, excessive contact pressure was the main failure mechanism of the rubber core [12,13,14]. Zhang et al. revealed the dynamic sealing mechanism of rubber through a tribological experiment of rubber and metal. They concluded that the contact pressure increased with increasing well pressure and friction coefficient [15]. Polonsky and Ma investigated the impact of contact pressure on packer seal performance, showing that higher contact pressure results in better seal performance [16,17].

Based on these above studies, most scholars have assessed the sealing performance of a BOP core by numerical simulations, but there is no report about the effect of BOP shell defects on sealing performance. First, this research carried out a statistical analysis of failure cases, as detailed in Section 2. Then, the cause of BOP shell key locations and types of seal failure were determined through uniaxial compression and tensile tests to select a disc with a nitrile rubber material constitutive model.

The parameters of the BOP analysis model were set, and a calculation method for seal failure contact was proposed. Finally, the sealing performance of a top seal structure with defects was analyzed by the finite element method. The distribution characteristics of contact pressure at the top seal parts with different levels of wear and corrosion were obtained and the failure index of the top seal was quantified. These research results are of great significance for improving the sealing performance of ram BOPs, which can effectively ensure the safe operation of BOPs, avoid accidents, and provide a new idea for the maintenance and remanufacturing of BOPs.

## 2. Statistical Analysis of BOP Failure Cases

When a ram BOP switches a well, high pressure hydraulic fluid enters the oil cavities of the two cylinders and pushes both the piston and piston rod, moving the left and right ram assemblies to the center or sides of the wellbore, respectively, along the tracks defined by the ram interior guide bars. The rubber on the upper end of the ram is in close contact with the top of the ram chamber of the blowout preventer to create a sealing effect. The working principle is shown in Figure 1.

Statistical analysis was conducted based on the 2017–2019 well control equipment annual report released by the U.S. Department of Transportation. The proportion of foreign BOP failure types is shown in Table 1. Subsea BOP faults accounted for 92.5%, 94.2%, and 91.2% of failures, with an average of 92.6%. Above-water BOP failures accounted for 7.5%, 5.8%, and 8.8%, with an average of 7.4%. The main failures of BOP equipment were wear and tear, maintenance errors, and design issues. The average failure rates of component wear were 53.6%, 56.5%, and 46.5%, respectively. Wear and tear, and design issues accounted for approximately 20%. Therefore, it is important to carry out quantitative diagnoses and evaluations of the key position of BOPs.

A ram BOP is mainly composed of a shell, side door, ram assembly, liquid cylinder assembly, and other components. As the shell is the main bearing part in the working process of the BOP, its performance is directly related to the bearing capacity of the BOP; other parts are wearing parts that can be replaced. Therefore, this paper mainly examines the failure case analysis of the ram BOP shell. At present, 263 maintenance cases of BOP shells have been collected and statistically analyzed. The distribution of vulnerable parts and failure types of ram BOP shells in failure cases is shown in Figure 1.

In Figure 2, it is apparent that for the BOP shell, the main defects are backing ring groove damage, top seal damage, wear of the main diameter, and threaded hole damage, accounting for 33.61%, 27.46%, 22.95%, and 9.02%, respectively. Since the top seal of a ram BOP is under the condition of reciprocating dynamic load, the damage of the top seal accounts for a high proportion of failure cases in ram BOP shells. According to the existing data, the main failure types of the top seal are corrosion and wear. A case of ram BOP top seal corrosion and wear damage is shown in Figure 3; top seal corrosion depth is 5 mm and wear depth is 1 mm, and the ram BOP is a type of 2FZ35/70. Therefore, it is necessary to quantitatively evaluate the corrosion and wear failure indices of the top seal to guarantee the safe and high-efficiency operation of BOPs.

## 3. Analysis of the Contact Seal Failure Evaluation Method for Ram BOP

In this paper, uniaxial tensile and compressive tests of nitrile butadiene rubber (NBR) used for the top seal of a ram BOP are carried out. Furthermore, an NBR constitutive model is optimized to characterize the stress–strain relationship of NBR during the closing process. The contact seal method is used to explore the damage rule of the top seal, study the influence of different corrosion and wear defect sizes on the top seal, and put forward the contact seal failure criterion of the ram BOP top seal.

### 3.1. Constitutive Relation Analysis of Rubber

The core material of ram preventer rubber is NBR. Deformation, wear, crack, and other defects are liable to occur under long-term use, as shown in Figure 4. The maximum length of wear is 62 mm. The reliable seal of the ram BOP is related to the performance of the rubber core itself. Nitrile rubber is a typical large-deformation, incompressible, nonlinear material, whose mechanical property characterization is complex. The Mooney–Rivilin model assumes that the material behaves as an incompressible, isotropic, hyperelastic solid, and it relates the strain energy density of the material to the deformation gradient tensor. Molecular statistics theory and phenomenological theory are typically used to describe the mechanical properties of rubber materials. Molecular theory infers that the microstructure of rubber material is a molecular chain network composed of flexible long-chain molecules with arbitrary orientation through sparse intermolecular crosslinking points. Due to the low intermolecular force, its stress-strain behavior mainly depends on conformational entropy [18]. When there is no external force, the conformational entropy of the molecular chain is close to the maximum value. However, under the action of external forces, the conformation number changes due to the rotational motion within the molecular chain, resulting in the change of conformational entropy, which makes rubber materials have high elasticity [19]. According to the theory of molecular statistics, the stress–strain behavior of the rubber material is mainly determined by the conformational entropy. The theory assumes that the core macromolecules are randomly oriented long chain molecules, and a cross-linking network structure is formed through chemical cross-linking at the nodes of the molecular chain. The phenomenological theory of rubber elasticity can be expressed by the strain energy function, which includes neo-Hookean, Mooney–Rivilin, Yeoh, Ogden, and other models [20,21,22,23].

The deformation of rubber material adopts phenomenological theory, which mainly solves the elastic deformation of rubber. Since rubber is a hyperplastic material, the constitutive relation of this kind of material can be expressed as a function of three invariants—$I_{1},I_{2},I_{3}$—of the deformation tensor by the strain energy function (*W*) or as a function of three principal elongation ratios—$\lambda_{1},\lambda_{2},\lambda_{3}$—i.e., [24]:(1)$$\left\{ \begin{array}{l} W=W\left(I_{1},I_{2},I_{3}\right)\\ I_{1}=\lambda_{1}{}^{2}+\lambda_{2}{}^{2}+\lambda_{3}{}^{2}\\ I_{2}=\lambda_{1}{}^{2}\lambda_{2}{}^{2}+\lambda_{2}{}^{2}\lambda_{3}{}^{2}+\lambda_{1}{}^{2}\lambda_{3}{}^{2}\\ I_{3}=\lambda_{1}{}^{2}\lambda_{2}{}^{2}\lambda_{3}{}^{2} \end{array}\right.$$

The common polynomial model of the strain potential energy function can be obtained from Equation (1), and the Taylor expansion of the strain energy function can be expressed as Equation (2):(2)$$W=\sum_{i+j=1}^{N}c_{ij}{\left(I_{1}-3\right)}^{i}{\left(I_{2}-3\right)}^{j}+\sum_{k=1}^{N}\frac{1}{D_{k}}{\left(J-1\right)}^{2k}$$
where N is the model order, and $C_{ij}$ is the shear performance parameter of the material. $D_{k}$ is the material compression property parameter; J is the elastic volume ratio.

The Equation (2) is simplified to obtain the Mooney–Rivilin model, whose strain energy function can be expressed as Equation (3):(3)$$W=\sum_{i+j=1}^{N}c_{ij}{\left(I_{1}-3\right)}^{i}{\left(I_{2}-3\right)}^{j}$$

If $C_{01}=0$ in the Mooney–Rivilin model, the neo-Hookean model can be obtained, and its strain energy function can be expressed as Equation (4):(4)$$W=C_{10}(I_{1}-3)$$

The Yeoh model is a special type of polynomial with N=3, and its strain energy density function is shown in Equation (5):(5)$$W=\sum_{i=1}^{3}C_{i0}\left(I_{1}-3\right)+\sum_{k=0}^{3}\frac{1}{D_{k}}{\left(J-1\right)}^{2k}$$

The strain energy density function of the Ogden model can be expressed as Equation (6):(6)$$W=\sum_{i=1}^{N}\frac{\mu_{i}}{\alpha_{i}}\left(\lambda_{1}^{\alpha_{i}}+\lambda_{2}^{\alpha_{i}}+\lambda_{3}^{\alpha_{i}}-3\right)+\sum_{k=1}^{N}\frac{1}{D_{k}}{\left(J-1\right)}^{2k}$$
where $\mu_{i}$ and $\alpha_{i}$ are material constants.

### 3.2. Uniaxial Compression and Tensile Testing of Nitrile Butadiene Rubber

According to Chinese standards GB/T7757-2009 and GB/T 528-2009, the nitrile rubber material is made into a cylinder for the uniaxial compression test, and the sample is made into a tensile test. The diameter of the cylinder is 29 ± 0.5 mm, and the height is 12.5 ± 0.5 mm. The length, width, and thickness of the dumbbell-shaped tensile specimen are 115 mm, 6 mm, and 2 mm, respectively. The equipment used in the experiment is an electronic universal testing machine with E43 microcomputer provided by the MTS Company. Its maximum test force is 5 kN. The test equipment and samples are shown in Figure 5. First, the sample was kept in a 90 °C test chamber for 30 min. Then, the rubber sample was placed into the center of the pressure plate of the compression machine. The sample was compressed at a speed of 10 mm/min until the strain reached 25%. The test sample was then relaxed at the same speed and repeated three times. The fourth time was the formal experiment. The purpose of this operation was to eliminate any experimental error caused by the Mullins effect of the rubber material. A constant rate of 500 ± 50 mm/min was used to draw the average of five data sets as the tensile test results.

To study the constitutive relationship of rubber materials in uniaxial tensile and compression tests, the fitting results shown in Figure 6 and Figure 7 can be obtained using the four different constitutive models mentioned above combined with the test data. It can be clearly seen in the figures that the Mooney–Rivilin model has the highest coincidence degree with the uniaxial compression curve. Therefore, the Mooney–Rivilin model was used to characterize the stress and strain relationship of rubber in the process of gate closing. The shear performance parameters of the material were obtained as C10 = 5.002 MPa and C01 = −3.661 MPa by using the Mooney-Rivilin model, which was taken into the finite element analysis as the material parameters of the rubber model.

### 3.3. Establishment of the BOP Model

The ram BOP was modeled using 3D solid units. The top seal structure was meshed by the 20-node hexahedral element and its corresponding degenerate element using ANSYS 19.0 software, and the finite element mesh number was 50682. The displacement component constraint method was used to simulate the spatial displacement restriction of ram ring groove on rubber ring. Constraints of the brake-ram cavity model: the Z-direction displacement constraint was applied to the surface on the contact side of the rubber ring and the ram cavity; radial constraints were applied on the inner and outer surfaces of the rubber ring; circumferential constraint was applied to the rubberized circumferential truncation surface. The sectional view of the geometric model and the grid partition diagram are shown in Figure 8. In this paper, the damage rule of the BOP is studied at a rated working level of 70 MPa.

The material parameters of the ram BOP shell and ram are shown in Table 2.

### 3.4. Methods for Failure Analysis of Contact Seals

The essence of sealing is to prevent mass exchange between the sealed space and the surrounding medium [24,25]. The top sealing connection structure of the ram BOP occurs mainly through the top sealing rubber core installed on the upper plane of the ram plate. In the presence of pressure in the wellbore, the top sealing rubber core presses the top surface of the shell to form a sealing surface to achieve the sealing effect. The sealing performance of the BOP directly determines its performance. Suppose the contact pressure between the rubber core and the ram cavity is small. In this case, the rubber core does not completely contact the ram cavity, or the contact surface is too small, resulting in liquid leakage being a common sealing failure problem in a BOP. Therefore, a method of contact sealing is proposed to evaluate the sealing performance between rubber material and metal [26,27]. Assuming that the average contact pressure of the contact seal of the flawless BOP is $F_{0}$, the average contact pressure of the BOP for defect i is $F_{i}$, and the average contact pressure decreased by $\mu_{i}$.
(7)$$\mu_{i}=\left(\frac{F_{i-1}}{Fi}-1\right)\times 100\%$$

According to the relevant evaluation methods of sealing performance:

If $\mu_{i}\geq 20\%$ or $\sum_{i=1}^{n}\mu_{i-1}\geq 20\%$, the ram BOP contact seal fails.If $\mu_{i}<20\%$ or $\sum_{i=1}^{n}\mu_{i-1}<20\%$, then the ram BOP contact seal is reliable and can be used continuously.

## 4. Analysis of the Influence Law of the BOP Contact Seal

### 4.1. Sealing Performance Analysis of Defect-Free BOP Top Seal

The sealing performance of the ram BOP top seal was analyzed by the finite element method. The contact unit was applied to the ram cavity–rubber ring contact area. Displacement component constraint was used to simulate the spatial displacement restriction of the ram ring groove on the rubber ring, and 70 MPa pressure was applied on the contact surface between the ram and the top seal. The convergence of the finite element model was weak due to the incompressibility of the core material, and the compression limit of the calculation model was reached after 1481 iterations. The displacement distribution of the structural glue core at the sealing connection part is shown in Figure 9a. The distribution of the glue core deformation is uniform, and the maximum deformation is 1.48 mm. The equivalent stress distribution of the structure at the top sealing connection is shown in Figure 9b. The maximum equivalent stress in the region is no more than 40 MPa, and the position appears in the interior of the shell on the ram cavity. Figure 9c shows the equivalent stress distribution in the contact area between the rubber core and the top of the ram cavity. The maximum equivalent stress is located at the upper part of the rubber core near the outer surface, and its value is 0.95 MPa. The overall stress distribution gradually decreases from top to bottom. Figure 9d shows the contact pressure distribution in the contact area. The maximum and average contact pressure values are 42.9 MPa and 41.6 MPa, respectively.

### 4.2. Sealing Performance Analysis of the BOP Top Seal with Corrosion Defects

The corrosion defect model of the top seal was established by a Boolean operation with a spherical diameter of 0.1 m. The depth of the corrosion defect was controlled by the relative position of the spherical entity to the ram cavity model. The model of the top seal structures with corrosion depths of 1.0 mm, 2.0 mm, 3.0 mm, 4.0 mm, 5.0 mm, 6 mm, and 7 mm were solved nonlinearly.

As seen in Figure 10, for the top sealing structure under a 70 MPa load, with the increase in corrosion depth, the size of the whole contact pressure is radially and circumferentially the same, the contact pressure near the location of the corrosion pit can produce a change, and the maximum contact pressure is on the edge of the corrosion pit location, with the increase in corrosion depth extending outward.

With minimum contact pressure on the top of the corrosion pit position, when the corrosion depth was greater than 5 mm, the top defect was close to the zero-contact pressure area. It is shown that with the reduction of contact pressure between the rubber core and gate chamber, although there was a complete, airtight touch between each other, it did not form a complete, airtight security seal area, which will lead to seal failure.

As shown in Figure 11, the contact pressure decreases as the defect depth increases. When the defect depth is less than or equal to 4 mm, the maximum contact pressure between the rubber core and the ram cavity slightly increases compared with the case without defects. The reason may be that the corrosion pit is small, and the maximum contact pressure rises due to the stress concentration at this position. When the defect is greater than 4 mm, the maximum contact pressure is lower than that without the defect. The contact pressure tends to be stable in the early stage and decreases rapidly in the late stage. The slope of the downward curve is large, increasing the risk of top seal failure.

The decreasing trend of contact pressure with increasing corrosion defect depth is shown in Figure 12. When the defect depth is less than 1 mm, the average contact pressure is 41.76 Mpa. The pressure is 0.4% higher than that in the case of no defect, at 41.6 MPa, indicating that defects smaller than 1 mm have no obvious influence on the sealing performance of the top seal structure. When the defect depth is 2 mm, 3 mm, and 4 mm, the average contact pressure changes steadily, which are 3.4%, 4.5%, and 6.7%, respectively, and their values are all less than 7%. When the defect depth is 5 mm and 6 mm, the average contact pressure is 31.9 MPa and 27.6 MPa, which decreases by 23.1% and 33.6%, respectively, compared with the maximum contact pressure without defects. When the defect depth is 5 mm, the average contact pressure between the core and the ram cavity decreases by 23.3%. According to Formula (7), the corrosion depth will lead to seal failure. Therefore, to ensure the safety of the ram BOP during operation, the critical failure size of the ram BOP corrosion depth is recommended to not exceed 4.0 mm, as seen in Figure 12.

### 4.3. Sealing Performance Analysis of the BOP Top Seal with Wear Defects

The top seal wear of ram BOP was simulated by the way that the arc-shaped surface of different depths was tangent to the bottom surface of ram cavity, and the depth of different wear defects was characterized by the curvature of the arc-shaped surface. The defect model of ram and ram cavity wear is shown in Figure 13.

The model of the top seal connection structures with overall wear depths of 0.1 mm, 0.2 mm, 0.3 mm, 0.4 mm, 0.5 mm, 0.6 mm, 0.8 mm, 1.0 mm, and 1.5 mm were solved nonlinearly.

Under a load of 70 MPa, the overall contact pressure of the top seal structure decreases linearly with increasing top seal wear depth, which can be clearly seen in Figure 14. When the wear depth reaches 1.5 mm, the contact pressure at the defect with the largest wear gradually decreases and finally approaches zero, leading to seal failure. The contact pressure at the maximum depth of the top seal wear defect is the smallest, the contact pressure at both sides of the rubber core is higher than that at the maximum depth of the defect, and the contact pressure decreases gradually along the maximum depth of the wear defect.

As shown in Figure 15, the contact pressure decreases significantly with increasing wear depth. Since the wear defect is a planar integral defect, no stress concentration will occur on the contact surface, so the maximum and minimum contact pressures do not fluctuate up and down compared with corrosion defects, and the decreased amplitude is stable. With increasing wear depth, the interference of the rubber core in the ram cavity decreases, which leads to a decrease in the contact pressure and finally seal failure. When the wear depth reaches 1.0 mm, the average contact pressure is 3.33 MPa, and the minimum contact pressure is 1.98 MPa. Compared with 41.6 MPa and 37.78 MPa without defects, the contact pressure is almost zero, which can no longer meet the sealing requirements. The wear defect depth has a great influence on the sealing performance of the top seal joint structure.

Figure 16 shows that when the defects wear, the maximum and average contact pressure present a downward trend with increasing defect depth. When the wear depth is less than 0.6 mm, the contact pressure is linear, and the drop rate remains at approximately 10%. The basic structure shows that the stability of the top seal joint can also maintain good contact. When the wear depth is 0.3 mm, the average contact pressure between the rubber core and the ram cavity decreases by 28.6%, which is more than 20%. According to Formula (7), the wear depth will lead to seal failure. To ensure the safety of the ram BOP during operation, according to engineering experience, it is recommended that the ram BOP wear critical failure size not exceed 0.2 mm, according to Figure 16.

## 5. Conclusions

This paper aims to address the problem that the top seal of the BOP is prone to corrosion and wear, resulting in seal failure. A geometric model of the top seal connection structure is established, and the damage law and reliability of its contact seal are studied. The following conclusions are drawn:(1)The data were obtained by uniaxial tensile and compression tests of NBR. The stress and strain curves of the four constitutive models of hyperplastic materials were fitted using the test data. The Mooney–Rivilin constitutive model was selected, and the Rivilin coefficients were calculated as C10 = 5.002 MPa and C01 = −3.661 MPa. The model can accurately characterize the nominal stress–strain relationship in the working process of the top seal rubber core.(2)The damage to the ram BOP top seal can be effectively characterized by the contact seal method. With the increase in the top seal corrosion pit depth, the overall contact pressure distribution of the top seal connection structure is uniform, and the local contact pressure at the edge of the corrosion pit will increase, while the contact pressure at most other positions of the corrosion pit will gradually decrease. With increasing top seal wear depth, the overall contact pressure decreases linearly. The contact pressure at the maximum depth of the wear defect of the top seal is the smallest, and the contact pressure decreases gradually along the maximum depth of wear.(3)The failure analysis method of contact seals for ram BOP top seals is proposed and the reliability evaluation problem of quantitative seals for ram BOP top seals is solved. When the top seal corrosion pit depth reached 5.0 mm, the average contact pressure decreased by 23.3% compared with that without defects. When the top seal wear depth reached 0.3 mm, the average contact pressure decreased by 28.6% compared with the non-defect condition, and the ram BOP top seal was judged as sealing failure. These research results are aimed at ensuring the safe operation of a ram BOP. It is suggested that the acceptable depth of the seal corrosion pit at the top of the ram BOP is 4.0 mm, and the acceptable depth of the wear is 0.2 mm.

## Figures and Tables

**Figure 1 materials-16-03413-f001:**
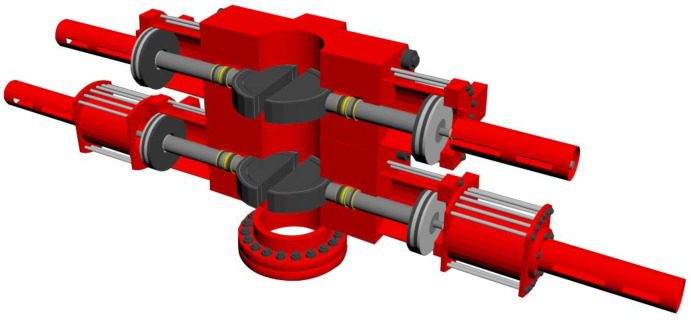
Schematic diagram of ram BOP working principle.

**Figure 2 materials-16-03413-f002:**
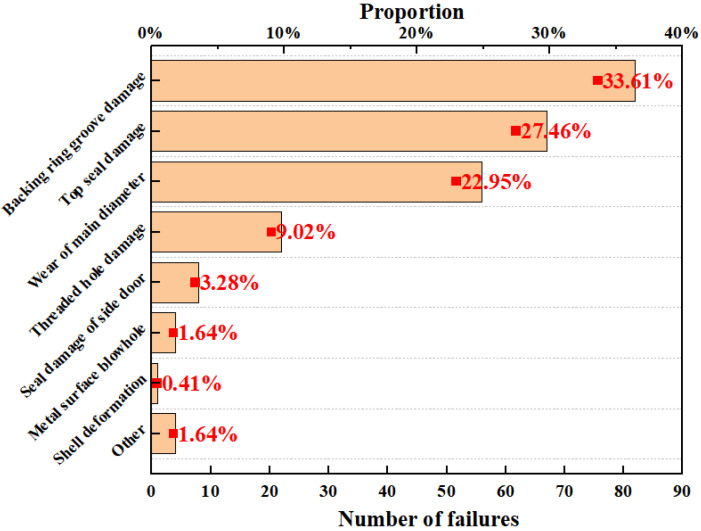
Proportion of main failure types of BOP shell.

**Figure 3 materials-16-03413-f003:**
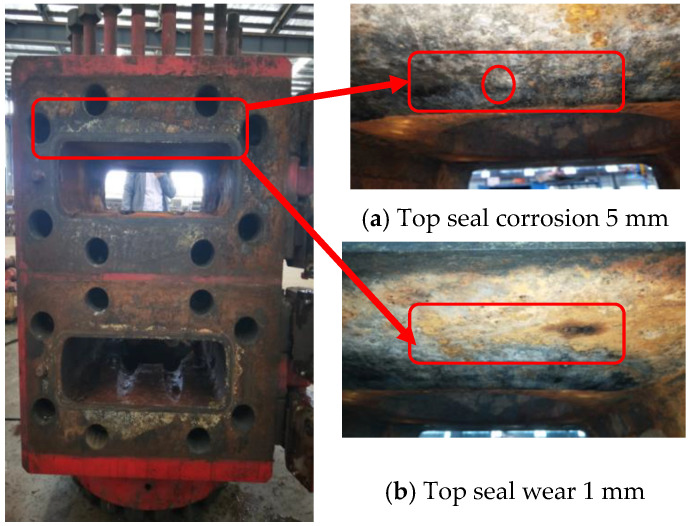
Top seal corrosion and wear defects.

**Figure 4 materials-16-03413-f004:**
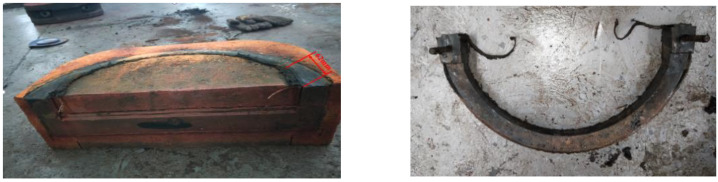
Rubber core failure.

**Figure 5 materials-16-03413-f005:**
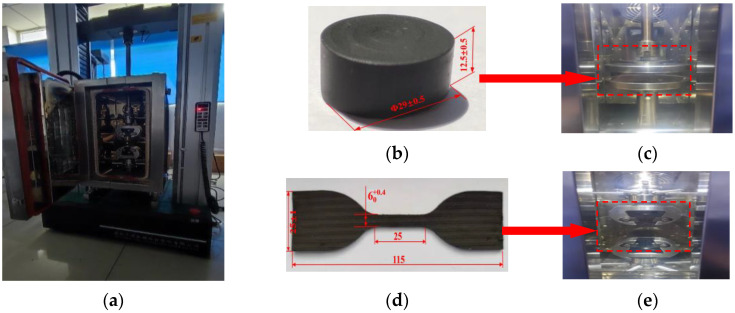
Test equipment and samples. (**a**) Mechanical property test equipment; (**b**) Single-axis compression test piece; (**c**) Uniaxial compression test; (**d**) Uniaxial tensile test piece; (**e**) Uniaxial tensile test.

**Figure 6 materials-16-03413-f006:**
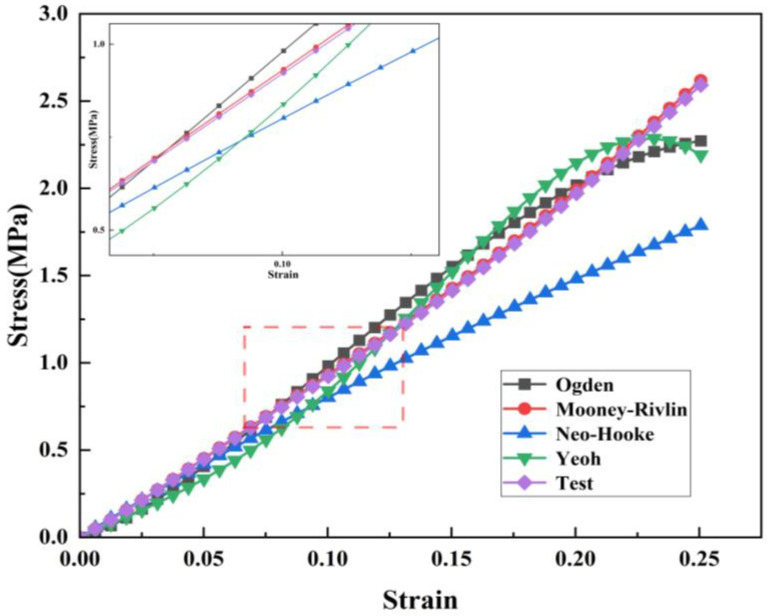
Compression test constitutive fitting curve.

**Figure 7 materials-16-03413-f007:**
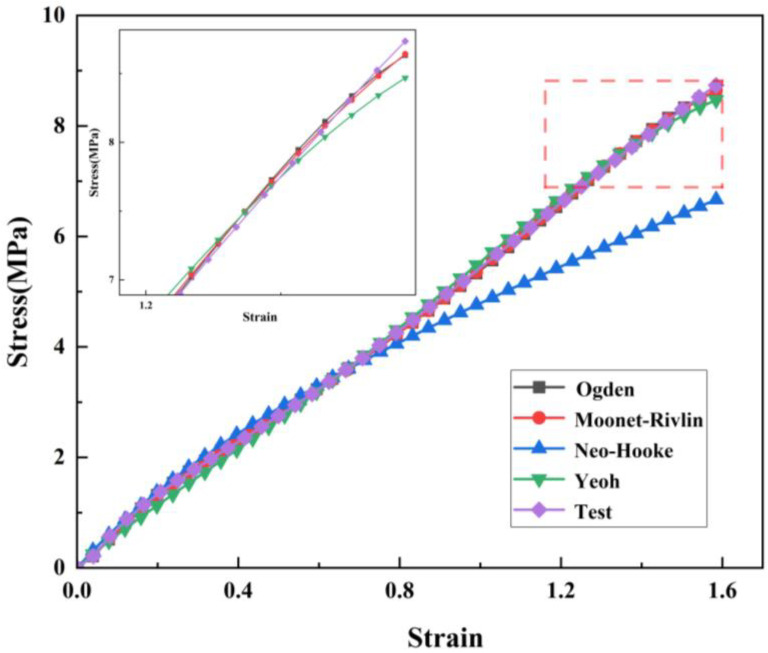
Tensile test constitutive fitting curve.

**Figure 8 materials-16-03413-f008:**
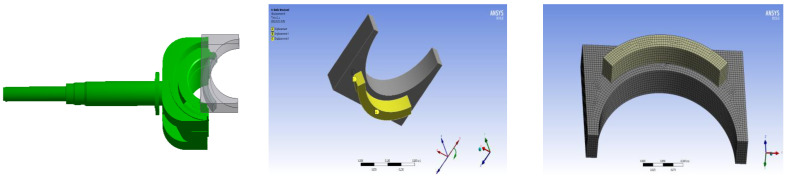
Model diagram of the BOP top seal.

**Figure 9 materials-16-03413-f009:**
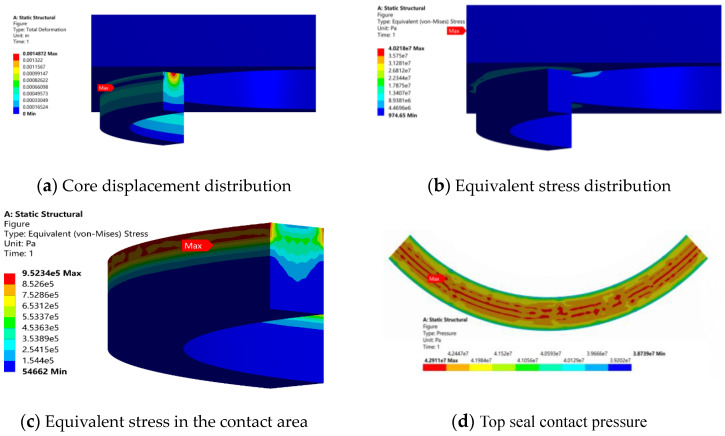
Analysis results of sealing performance of top seal of defect-free BOP.

**Figure 10 materials-16-03413-f010:**
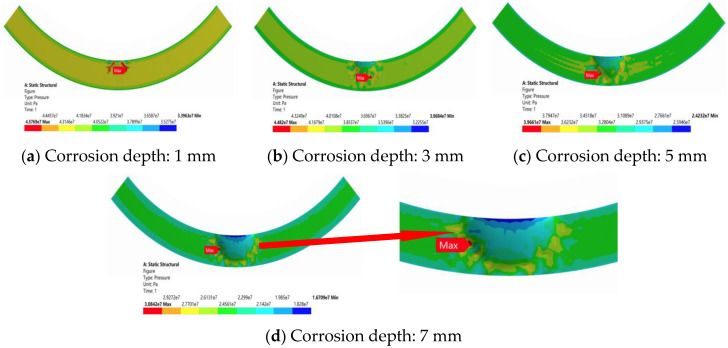
Contact pressure cloud diagram of the lower top seal structure with different corrosion defects.

**Figure 11 materials-16-03413-f011:**
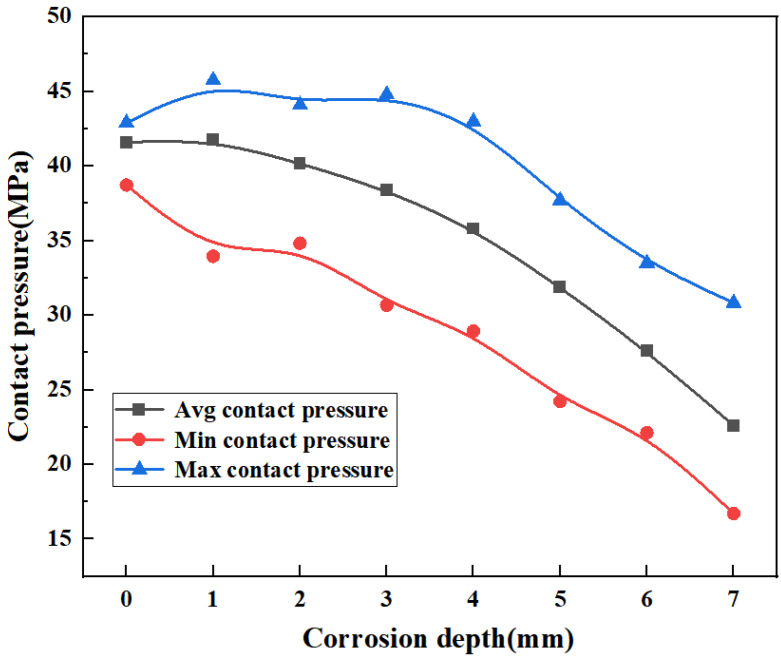
Top seal contact pressure with corrosion defects.

**Figure 12 materials-16-03413-f012:**
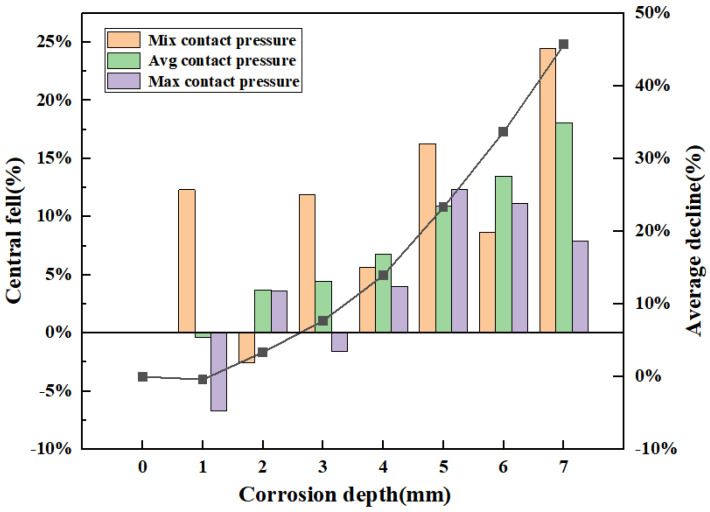
Downward trend of contact pressure in top seal structures with corrosion defects.

**Figure 13 materials-16-03413-f013:**
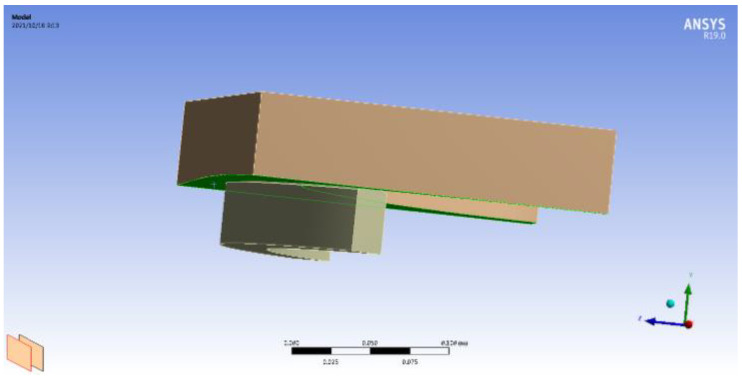
Defect model of ram and ram cavity wear.

**Figure 14 materials-16-03413-f014:**
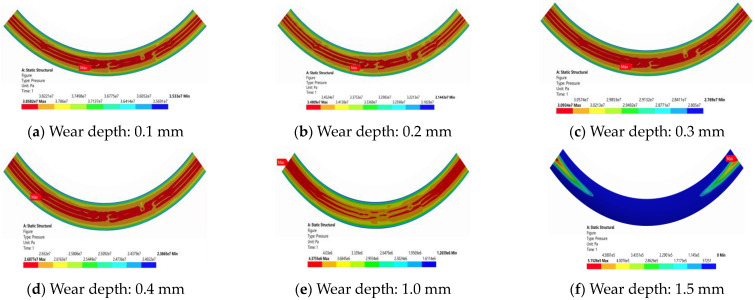
Contact pressure cloud diagram of the lower top seal structure with different wear defects.

**Figure 15 materials-16-03413-f015:**
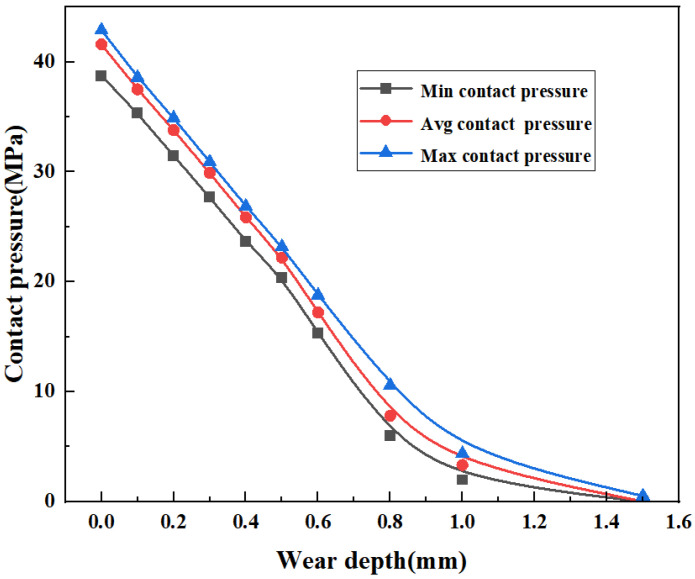
Contact pressure of the top seal with wear defects.

**Figure 16 materials-16-03413-f016:**
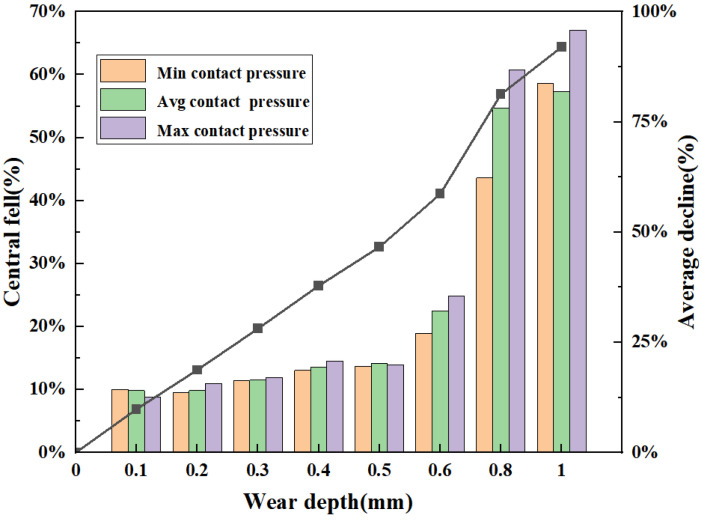
Downward trend of the contact pressure of the top seal structure with wear defects.

**Table 1 materials-16-03413-t001:** Proportion of foreign BOP failure types.

Statistical Year	BOP Fault Model/Ratio			Underwater BOP Fault Model/Ratio		
Failure Type	Wear and Tear	Maintenance Error	Design Issue	Wear and Tear	Maintenance Error	Design Issue
2017	53.6%	12.5%	7%	57.7%	13.5%	8.3%
2018	56.5%	7.2%	7.2%	52.4%	9.5%	14.3%
2019	46.5%	27.9%	7%	46.9%	20%	11.1%

**Table 2 materials-16-03413-t002:** Mechanical properties of materials of various parts.

Part	Material	Tensile Strength (MPa)	Yield Strength (MPa)	Modulus of Elasticity (GPa)	Poisson’s Ratio	Shrinkage (%)	Elongation (%)
Ram	40CrNiMo	1050	960	210	0.295	53	13
Shell	25CrNiMo	724	438	214	0.28	54	22

## Data Availability

Not applicable.

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
