# Peer review of "Influence of Top Seal Damage on Contact Seal in Ram Blowout Preventer"

_materials, 2023, doi:10.3390/ma16093413_

Round 1
Reviewer 1 Report
In this paper, a calculation method is proposed to quantitatively evaluate BOP seal contact failure.Sealing performance of top seal structure with defects was analyzed by finite element method. The results of these studies are considered important to avoid accidents and improve BOP sealing performance.However, since the description and explanation are insufficient, it is difficult for the reader to understand.Please improve the following points.
(1)In order to understand the parts such as rubber packing, the direction of movement of each part, and the direction of force action, diagrams to explain the mechanism of the BOP are necessary. Describe the problem setting more precisely. Many readers are not familiar with this subject.
(2)If you abbreviate "blowout preventer" to BOP, be consistent throughout your paper.
(3)If dimensions are required in all figures such as Figures 2 and 3, please enter them.
(4)Section 3.3, Fig. 7: Enter the name of each part. There is no mesh division diagram, but how did you set the mesh size? Enter the total number of elements and total number of nodes. What software did you use?
(5)Equation (7): The use of the subscript i assumes the existence of multiple defects. If so, isn't the summation sign necessary?
(6)Section 4.1: Figure 8 is blurry and small, which is unfriendly to the reader.
(7)Figs. 11 and 13:Why does contact pressure decrease due to increased corrosion depth and wear?Which part receives the relieved pressure instead?
Reviewer 2 Report
The paper investigates the influence of top seal damage on contact seal in ram blowout preventer which is one of the main factors leading to well control risk.The study concerns the exploration and development of oil and gas resources and the harsh conditions of drilling operations where spillovers may occur. Accidents in this area can cause human casualties and serious environmental pollution. In this sense, the research proposed and the conclusions drawn are relevant and important for the modern industry.
I have one requirement for the authors. It is possible to give more detailed data on the loading of the model with the method of finite elements (Fig. 8)?
The paper is well written and properly formatted and can be published in present form.
Reviewer 3 Report
The paper presents an FEM approach on the evaluation of sealing performance of a ram blowout preventer
The paper needs some clarifications, mainly on defining the evaluation method:
Line 170 and Figures 5 and 6 – Please check the strain value. Is it 25% which means 0.25, or is it 0.25% like on the figures scale?
Line 168 – Why this temperature? Is 90C degrees the temperature at which the sealing is working? Is the rubber tested, the same with the rubber used for sealing?
At least two load cases are presented in Figure 8. Which are these?
Is the seal corrosion presented in Fig. 2 related with the corrosion analysis from paragraph 4.2?
Is the ram BOP from Fig. 2 the same with the one from your FEM analysis?
A description of the Ram BOP model must be added, maybe related with the FEM model. Where is the reciprocating movement?
The Loads and Constraints on the FEM model must be explained. A picture is useful.
It is not clear how the wear is modeled. A picture is useful.
Reviewer 4 Report
In their work, the authors presented the results of research on the impact of the wear of the actuator sealant preventing blowouts on the effectiveness of the contact seal. The subject matter of the article is interesting and up-to-date. As the authors rightly pointed out in Chapter 1, most of the research on this subject does not take into account the impact of the condition of the sealed surface on the effectiveness of sealants. This confirms the need to conduct research, the results of which are presented in the article.
As part of the conducted research, the authors diagnosed the most common causes of sealant wear, and then developed a numerical model that allows for a virtual mapping of material properties and sealant operating conditions. It should be noted, however, that the developed numerical model has not been fully verified. This makes it difficult to make a reliable assessment of the results obtained.
Other important notes to the manuscript:
1. To better illustrate the analyzed problem, it is necessary to add a diagram that will show the way the sealant works and loads.
2. During operation, the sealant is subjected not only to tensile and compressive loads, but also to shear. Has the effect of shear been taken into account when determining material properties?
3. It is required to extend the description of the numerical model that was used for the simulation. First of all, there is no diagram of the developed geometric model with an indication of the method of fixing and loading the sealant, there is also no detailed description of the boundary conditions adopted for the calculations.
4. The research was based solely on the results of numerical modeling, which were not verified in experimental studies. Were studies carried out that would allow to compare the obtained results with real values?
The work needs to be extended with experimental verification
Reviewer 5 Report
1. The novelty of this work needs to be elaborated more. Why this study is so important? How it differentiates from the other methods, e.g.
http://dx.doi.org/10.1016/j.engfailanal.2015.08.025
2. The abstract is not well done because it lists test results instead of showing the work approach and steps, a description of the objectives, a detailed presentation of the steps and means to reach the objectives as well as a list of the results that’s will be discussed further.
3. Add references for the mentioned data in Table 1.
4. References for the maintenance cases mentioned in line 97 should be included.
5. Add more details about 2FZ35/70.
6. Authors have not mentioned any methods for pitting corrosion prevention.
7. Can you explain if there is a connection between the case presented in Figure 2 and the numerical or experimental investigation?
8. Add more details and references for the theory of molecular.
9. The Mooney–Rivilin model is based on the assumption that the material behaves like an incompressible, isotropic, hyperelastic solid, and it relates the strain energy density of the material to the deformation gradient tensor. You should add this assumption with this model.
10. Add the differences between the Neo-Hookean and Mooney-Rivlin models and the advantages of the second model.
11. Have you discussed the influence of strain rate or temperature on the mechanical properties of NBR.
12. The numerical and experimental parts are not rigorous and is missing information that are crucial to the reproduction of the proposed experimental and numerical simulations. (Used finite element software with further details).
13. The reasons behind the high values of von Mises stress observed in Figure 8 need to be explained.
14. Could you provide more details on the methods and techniques used to measure the corrosion depth and wear depth in the study represented in figures 11 and 14?
Round 2
Reviewer 3 Report
The paper has been improved according to requests
Paper good for publishing
Author Response
Dear reviewer
Many thanks to the review experts for their valuable comments on the article.
Thank you very much.
Sincerely
Reviewer 4 Report
Thank you very much for the comprehensive answers to the questions. However, I still have some insufficiency related to the lack of experimental verification. In your publication, you rely on the results of experimental research related to the amount of consumption. However, I have no reference to research confirming the results of the FEM analysis. The actual amount of consumption was used as boundary conditions for the calculations. However, it is difficult to refer to the quality of the results obtained later.
Author Response
Dear reviewer
Many thanks to the review experts for their valuable comments on the article. This article mainly studies the Influence of top seal damage on contact seal in ram blowout preventer. The finite element analysis in this paper is to prevent the failure case of the injector as a reference. The conclusions of the study serve as recommendations for the maintenance of the blowout preventer. In the later research process, it will be combined with the actual engineering application situation to further improve and demonstrate.
Thank you very much.
Sincerely
Reviewer 5 Report
The majority of my concerns have been adequately addressed by the authors.
Author Response

(The authors gave the same response as above.)
